# DISCOVERY OF PREDICTIVE REPRESENTATIONS WITH A NETWORK OF GENERAL VALUE FUNCTIONS

## ABSTRACT

The ability of an agent to *discover* its own learning objectives has long been considered a key ingredient for artificial general intelligence. Breakthroughs in autonomous decision making and reinforcement learning have primarily been in domains where the agent's goal is outlined and clear: such as playing a game to win, or driving safely. Several studies have demonstrated that learning extramural sub-tasks and auxiliary predictions can improve (1) single human-specified task learning, (2) transfer of learning, (3) and the agent's learned representation of the world. In all these examples, the agent was instructed what to learn about. We investigate a framework for discovery: curating a large collection of predictions, which are used to construct the agent's representation of the world. Specifically, our system maintains a large collection of predictions, continually pruning and replacing predictions. We highlight the importance of considering stability rather than convergence for such a system, and develop an adaptive, regularized algorithm towards that aim. We provide several experiments in computational micro-worlds demonstrating that this simple approach can be effective for discovering useful predictions autonomously.

## 1 INTRODUCTION

The idea that an agent's knowledge might be represented as predictions has a long history in machine learning. The first references to such a predictive approach can be found in the work of Cunningham (1972), Becker (1973), and Drescher (1991) who hypothesized that agents would construct their understanding of the world from interaction, rather than human engineering. These ideas inspired work on predictive state representations (PSRs), as an approach to modeling dynamical systems. Simply put, a PSR can predict all possible interactions between an agent and it's environment by reweighting a minimal collection of core test (sequence of actions and observations) and their predictions, as an alternative to keeping a finite history or learning the one step latent dynamics of the world, as in a POMDP. Extensions to high-dimensional continuous tasks have demonstrated that the predictive approach to dynamical system modeling is competitive with state-of-the-art system identification methods (Hsu et al., 2012). One important limitation of the PSR formalism is that the agent's internal representation of state must be composed exclusively of predictions.

Recently, Sutton et al. (2011) introduced a formalism for specifying and learning large collections of predictions using value functions from reinforcement learning. These General Value Functions (GVFs), can represent a wide array of multi-step state contingent predictions (Modayil et al., 2014), while the predictions made by a collection of GVFs can be used to construct the agent's state representation (Schaul and Ring, 2013; White, 2015). State representation's constructed from predictions have been shown to be useful for reward maximization tasks (Rafols et al., 2005; Schaul and Ring, 2013), and transfer learning (Schaul et al., 2015). One of the great innovation of GVFs is that we can clearly separate (1) the desire to learn and make use of predictions in various ways, from (2) the construction of the agent's internal state representation. For example, the UNREAL learning system (Jaderberg et al., 2016), learns many auxiliary tasks (formalized as GVFs) while using an actor-critic algorithm to maximize a single external reward signal—the score in an Atari game. The auxiliary GVFs and the primary task learner share the same deep convolutional network structure. Learning the auxiliary tasks results in a better state representation than simply learning the main task alone. GVFs allow an agent to make use of both increased representational power of predictive representations, and the flexibility of state-of-the-art deep learning systems.

In all the works described above, the GVFs were manually specified by the designer; an autonomous agent, however, must discover these GVFs. Most work on discovery has been on the related topics of temporal difference (TD) networks (Makino and Takagi, 2008) and options (Konidaris et al., 2011; Mann et al., 2015; Mankowitz et al., 2016; Vezhnevets et al., 2016; Daniel et al., 2016; Bacon et al., 2017). Discovery for options is more related than TD networks, because similarly to a GVF, an option (Sutton et al., 1999) specifies small sub-tasks within an environment. Option discovery, however, has been largely directed towards providing temporally abstract actions, towards solving the larger task, rather than providing a predictive representation. For example, Bacon et al. (2017) formulated a gradient descent update on option parameters—policies and termination functions—using policy gradient objectives. The difficulty in extending such gradient-based approaches is in specifying a suitable objective for prediction accuracy, which is difficult to measure online.

We take inspiration from ideas from representation search methods developed for neural networks, to tackle the daunting challenge of GVF discovery for predictive representations. Our approach is inspired by algorithms that search the topology space of neural networks. One of the first such approaches, called the cascade correlation network learning typifies this approach (Fahlman and Lebiere, 1990). The idea is to continually propose new hidden units over time, incrementally growing the network to multiple levels of abstraction. To avoid the computation and memory required to pose units whose activation is de-correlated with the network activations, Sutton and Whitehead (1993) empirically demonstrated that simply generating large numbers of hidden units outperformed equal sized fixed networks in online supervised learning problems. Related approaches demonstrated that massive random representations can be highly effective (Rahimi and Recht, 2009; Andoni et al., 2014; Giryes et al., 2015). This randomized feature search can be improved with the addition of periodic pruning, particularly for an incremental learning setting.

In this paper, we demonstrate such a curation framework for GVF discovery, with simple algorithms to propose and prune GVFs. To parallel representation search, we need both a basic functional form—a GVF primitive—for each unit in the network and an update to adjust the weights on these units. We propose a simple set of GVF primitives, from which to randomly generate candidate GVFs. We develop a regularized updating algorithm, to facilitate pruning less useful GVFs, with a stepsize adaptation approach that maintains stability in the representation. We demonstrate both the ability for the regularizer to prune less useful GVFs—and the corresponding predictive features—as well as utility of the GVF primitives as predictive features in several partially observable domains. Our approach provides a first investigation into a framework for curation of GVFs for predictive representations, with the aim to facilitate further development.

## 2 General Value Function Networks

The setting considered in this paper is an agent learning to predict long-term outcomes in a partially observable environment. The dynamics are driven by an underlying Markov decision process (MDP), with potentially (uncountably) infinite state-space $\mathcal{S}$ and action-space $\mathcal{A}$, and transitions given by the density $P : \mathcal{S} \times \mathcal{A} \times \mathcal{S} \to [0, \infty)$. The agent does not see the underlying states, but rather only observes observation vector $\mathbf{o}_t \in \mathcal{O} \subset \mathbb{R}^m$ for corresponding state $\mathbf{s}_t \in \mathcal{S}$. The agent follows a fixed behaviour policy, $\mu : \mathcal{S} \times \mathcal{A} \to [0, \infty)$, taking actions according to $\mu(\mathbf{s}_t, \cdot)$. Though we write this policy as a function of state, the behaviour policy is restricted to being a function of input observations—which is itself a function of state—and whatever agent state the agent constructs. The agent aims to build up a predictive representation, $\mathbf{p}_t = f(\mathbf{o}_t, \mathbf{p}_{t-1}) \in \mathbb{R}^n$ for some function $f$, as a part of its agent state to overcome the partial observability.

GVF networks provide such a predictive representation, and are a generalization on PSRs and TD networks. A General Value Function (GVF) consists of a target policy $\pi : \mathcal{S} \times \mathcal{A} \to [0, \infty)$, discount function $\gamma : \mathcal{S} \times \mathcal{A} \times \mathcal{S} \to [0, 1]$ and cumulant $r : \mathcal{S} \times \mathcal{A} \times \mathcal{S} \to \mathbb{R}$. The value function $v_\pi : \mathcal{S} \to \mathbb{R}$ is defined as the expected return, $G_t$, where

$$G_t \stackrel{\text{def}}{=} r(\mathbf{s}_t, a_t, \mathbf{s}_{t+1}) + \gamma(\mathbf{s}_t, a_t, \mathbf{s}_{t+1})G_{t+1}.$$

A GVF prediction, for example, could provide the probability of hitting a wall, if the agent goes forward, by selecting a policy that persistently takes the forward action, a cumulant that equals 1 when a wall is hit and 0 otherwise and a discount of 1, until termination. We direct the reader to prior work detailing the specification and expressiveness of GVFs (Sutton et al., 2011); we also provide further examples of GVFs throughout this paper.

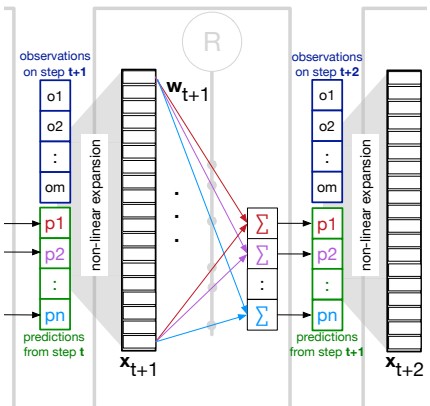

Figure 1: The inputs—observations $\mathbf{o}_{t+1}$ and GVF predictions $\mathbf{p}_t$ from the last time step—pass through a nonlinear expansion, such as a fixed neural network or tile coding, producing the feature vector $\mathbf{p}_{t+1}$. The feature vector is weighted linearly to produce the next set of predictions $\mathbf{p}_{t+1}$. This approach decouples the specification of the representation for the learner, which consist of both observations and predictive features, and the updating algorithm. Such a framework could be modified to include a history of observations; for simplicity here, we only consider using predictions to overcome partial observability and do not maintain histories of observations.

A GVF network consists of a set of GVFs $\{(\pi_i, \gamma_i, r_i)\}_{i=1}^n$ where the prediction vector $\mathbf{p}_t \in \mathbb{R}^n$ consists of the approximate value functions $\mathbf{p}_t = [\hat{v}_{\pi_1}(\mathbf{s}_t), \ldots, \hat{v}_{\pi_n}(\mathbf{s}_t)]$ and is computed as a function of the current observation and predictions of the previous step $\mathbf{p} = f(\mathbf{o}_t, \mathbf{p}_{t-1})$. This network is depicted in Figure 1. GVF networks can encode both PSRs and TD networks, providing a general formalism for predictive representations; for space, we more thoroughly describe the relationships to these approaches in Appendix A.

There has been some work towards the goal of learning $f$ to provide predictions $\mathbf{p}_t$, in the original papers on TD networks and TD networks with options and with a following algorithm using recurrent gradient updates (Silver, 2012). Additionally, there has been quite a bit of work in off-policy learning, with gradient temporal difference (GTD) algorithms (Sutton et al., 2009; Maei, 2011), which can be used to estimate the value functions for a GVF policy $\pi$, from experience generated by a different behaviour policy $\mu$. We leverage these works, to develop a new updating approach for GVF networks in the next section.

## 3   STABILITY VERSUS CONVERGENCE

When learning in a partially observable domain, the agent inherently faces a non-stationary problem. From the agent's perspective, a similar observation is observed, but the target may vary significantly, because hidden variables are influencing the outcome. For such settings, tracking has been shown to be critical for accuracy (Sutton et al., 2007), even when the underlying distribution is in fact stationary. Tracking—continually updating the weights with recent experience—contrasts the typical goal of convergence; much of the previous algorithm development, however, has been towards the aim of convergence (Silver, 2012).

We propose treating the learning system as a dynamical system—where the weight update is based on stochastic updates known to suitably track the targets—and consider the choice of stepsize as the inputs to the system to maintain *stability*. Such updates have been previously considered under adaptive gain for least-mean squares (LMS) (Benveniste et al., 1990, Chapter 4) (Sutton, 1992), where weights are treated as state following a random drift. These approaches, however, are designed particularly for the LMS update and so do not extend to the off-policy temporal difference learning algorithms needed to learn GVFs. To generalize this idea to other incremental algorithms, we propose a more general criteria based on the magnitude of the update.

Consider a generic update

$$\mathbf{w}_{t+1} = \mathbf{w}_t + \alpha \Delta_t \tag{1}$$

where $\Delta_t \in \mathbb{R}^d$ is the update for this step, for weights $\mathbf{w}_t \in \mathbb{R}^d$ and constant stepsize $\alpha$. Typically the update includes a normalization constant $c_t$, dependent on the norm of the features and target. For example, for normalized LMS predicting target $y_t$ from observation vector $\mathbf{o}_t$, $\Delta_t = \mathbf{o}_t(y_t - \mathbf{o}_t^\top \mathbf{w}_t)$ and $c_t = \|\mathbf{o}_t^\top \mathbf{o}_t\|_2^2 +$ an estimate of the variance of the noise in the targets. Such normalization ensures the update appropriately reflects descent direction, but is invariant to scale of the features and targets. The weights $\mathbf{w}_t$ evolve as a function of the previous weights, with stepsize $\alpha$ acting as a control input for how this system evolves.

A criteria for $\alpha$ to maintain stability in the system is to keep the norm of the update small

$$\min_{\alpha \geq \epsilon} \mathbb{E}\left[ \|\Delta_t(\alpha)\|_2^2 \mid \mathbf{w}_0 \right] \tag{2}$$

for a small $\epsilon > 0$ that provides a minimum stepsize. The update $\Delta_t(\alpha)$ on this time step is dependent on the stepsize $\alpha$, because that stepsize influences $\mathbf{w}_t$ and past updates. The expected value is over all possible update vectors $\Delta_t(\alpha)$, for the given stepsize and assuming the system started in some $\mathbf{w}_0$. If $\alpha$ is small enough to ensure updates are bounded, and policy $\pi$ and MDP satisfy the standard requirements for ergodicity, a stationary distribution exists, with $\Delta_t(\alpha)$ not dependent on the initial $\mathbf{w}_0$ and instead only driven by the underlying state dynamics and target for the weights.

In the next sections, we derive an algorithm to estimate $\alpha$ for this dynamical system, first for a general off-policy learning update and then when adding regularization. We call this algorithm AdaGain: Adaptive Gain for Stability.

### 3.1 ADAPTIVE GAIN FOR STABILITY WITH FIXED FEATURES

We generically consider an update $\Delta_t$ in (1) that includes both TD and GTD. Before deriving the algorithm, we demonstrate concrete updates for the stepsize. For TD(0), the update is

$$\delta_t \overset{\text{def}}{=} r_{t+1} + \gamma_{t+1}\mathbf{x}_{t+1}^\top \mathbf{w}_t - \mathbf{x}_t^\top \mathbf{w}_t \qquad \text{with } \gamma_t \overset{\text{def}}{=} \gamma(S_t, A_t, S_{t+1})$$

$$\Delta_t \overset{\text{def}}{=} c_t^{-1}\delta_t\mathbf{x}_t \qquad \text{with } c_t \overset{\text{def}}{=} \|\mathbf{x}_t\|_2^2 + 1$$

$$\alpha_t = (\alpha_{t-1} + \bar{\alpha}c_t^{-1}\langle \Delta_t, \mathbf{x}_t\rangle\langle \mathbf{d}_t, \boldsymbol{\psi}_t\rangle)_\epsilon \qquad \text{with } \mathbf{d}_t \overset{\text{def}}{=} \mathbf{x}_t - \gamma_{t+1}\mathbf{x}_{t+1}$$

$$\boldsymbol{\psi}_{t+1} = (1 - \beta)\boldsymbol{\psi}_t + \beta c_t^{-1}\left(\delta_t - \alpha_t(\mathbf{x}_t^\top \boldsymbol{\psi}_t)\right)\mathbf{x}_t$$

where $\alpha_0 = 1.0$, $\boldsymbol{\psi}_1 = \mathbf{0}$, $\bar{\alpha}$ is a meta-stepsize and $\beta$ is a forgetting parameter to forget old gradient information (e.g., $\beta = 0.01$). The operator $(\cdot)_\epsilon$ thresholds any values below $\epsilon > 0$ to $\epsilon$ (e.g., $\epsilon = 0.001$), ensuring nonzero stepsizes.

Another canonical algorithm for learning value functions is GTD($\lambda$), with trace parameter $\lambda_t \overset{\text{def}}{=} \lambda(S_t, A_t, S_{t+1})$ for a trace function $\lambda : \mathcal{S} \times \mathcal{A} \times \mathcal{S} \to [0, 1]$.

$$\mathbf{e}_t \overset{\text{def}}{=} \rho_t(\lambda_t\gamma_t\mathbf{e}_{t-1} + \mathbf{x}_t) \qquad \text{with importance sampling correction } \rho_t \overset{\text{def}}{=} \frac{\pi(S_t, A_t)}{\mu(S_t, A_t)}$$

$$\mathbf{h}_{t+1} = \mathbf{h}_t + \alpha_t^{(h)}c_t^{-1}(\delta_t\mathbf{e}_t - (\mathbf{x}_t^\top \mathbf{h}_t)\mathbf{x}_t)$$

$$\Delta_t = c_t^{-1}\left[\delta_t\mathbf{e}_t - \gamma_{t+1}(1 - \lambda_{t+1})\mathbf{x}_{t+1}(\mathbf{e}_t^\top \mathbf{h}_t)\right]$$

$$\alpha_t = \left(\alpha_{t-1} + \bar{\alpha}c_t^{-1}\langle \Delta_t, \mathbf{e}_t\rangle\langle \mathbf{d}_t, \boldsymbol{\psi}_t\rangle + \bar{\alpha}c_t^{-1}\gamma_{t+1}(1 - \lambda_{t+1})\langle \Delta_t, \mathbf{x}_{t+1}\rangle\langle \mathbf{e}_t, \bar{\boldsymbol{\psi}}_t\rangle\right)_\epsilon \tag{3}$$

$$\boldsymbol{\psi}_{t+1} = ((1 - \beta)\mathbf{I} - \beta\alpha_t c_t^{-1}\mathbf{e}_t\mathbf{d}_t^\top)\boldsymbol{\psi}_t - \beta\alpha_t c_t^{-1}\gamma_{t+1}(1 - \lambda_{t+1})\mathbf{x}_{t+1}\mathbf{e}_t^\top \bar{\boldsymbol{\psi}}_t + \beta\Delta_t$$

$$\bar{\boldsymbol{\psi}}_{t+1} = (1 - \beta)\bar{\boldsymbol{\psi}}_t - \beta\alpha_t^{(h)}c_t^{-1}\mathbf{e}_t\mathbf{d}_t^\top \boldsymbol{\psi}_t - \beta\alpha_t^{(h)}c_t^{-1}\mathbf{x}_t\mathbf{x}_t^\top \bar{\boldsymbol{\psi}}_t$$

For the auxiliary weights $\mathbf{h}_t$—which estimate a part of the GTD objective—we use a small, fixed stepsize $\alpha_t^{(h)} = 0.01$, previously found to be effective (White and White, 2016).

We consider the derivation more generally for such temporal difference methods, where both TD and GTD arise as special cases. Consider any update of the form

$$\Delta_t = c_t^{-1}\left[\delta_t\mathbf{e}_t + \mathbf{u}_t(\mathbf{e}_t^\top \mathbf{h}_t)\right]$$

for vectors $\mathbf{e}_t, \mathbf{u}_t$ not dependent on $\mathbf{w}_t$. For GTD($\lambda$), $\mathbf{u}_t = -\gamma_{t+1}(1 - \lambda_{t+1})\mathbf{x}_{t+1}$. We minimize (2) using stochastic gradient descent, with gradient for one sample of the norm of the update

$$\frac{\frac{1}{2}\partial\|\Delta_t(\alpha)\|_2^2}{\partial\alpha} = \Delta_t(\alpha)\frac{\Delta_t(\alpha)}{\partial\alpha}$$

$$= c_t^{-1}\Delta_t(\alpha)^\top \left[\mathbf{e}_t\frac{\partial\delta_t(\alpha)}{\partial\alpha} + \mathbf{u}_t\left(\mathbf{e}_t^\top \frac{\partial\mathbf{h}_t}{\partial\alpha}\right)\right].$$

We can compute the gradient of the TD-error, using

$$\frac{\partial}{\partial \alpha} \delta_t = \frac{\partial}{\partial \alpha} (r_{t+1} + \gamma_{t+1} \mathbf{x}_{t+1}^\top \mathbf{w}_t - \mathbf{x}_t^\top \mathbf{w}_t)$$

$$= (\gamma_{t+1} \mathbf{x}_{t+1}^\top - \mathbf{x}_t^\top)^\top \frac{\partial \mathbf{w}_t}{\partial \alpha}$$

where $\mathbf{d}_t \overset{\text{def}}{=} \gamma_{t+1} \mathbf{x}_{t+1}^\top - \mathbf{x}_t^\top$ and we can recursively define $\frac{\partial \mathbf{w}_t}{\partial \alpha}$ as

$$\boldsymbol{\psi}_t \overset{\text{def}}{=} \frac{\partial \mathbf{w}_t}{\partial \alpha} = \frac{\partial (\mathbf{w}_{t-1} + \alpha \Delta_{t-1}(\alpha))}{\partial \alpha}$$

$$= \frac{\partial \mathbf{w}_{t-1}}{\partial \alpha} + \alpha \frac{\partial \Delta_{t-1}(\alpha)}{\partial \alpha} + \Delta_{t-1}(\alpha)$$

$$= \boldsymbol{\psi}_{t-1} - \alpha c_{t-1}^{-1} \mathbf{e}_{t-1} \mathbf{d}_{t-1}^\top \boldsymbol{\psi}_{t-1} + \alpha c_{t-1}^{-1} \mathbf{u}_t \mathbf{e}_{t-1}^\top \frac{\partial}{\partial \alpha} \mathbf{h}_{t-1} + \Delta_{t-1}(\alpha)$$

We obtain a similar such recursive relationship for $\frac{\partial}{\partial \alpha} \mathbf{h}_{t-1}$

$$\bar{\boldsymbol{\psi}}_t \overset{\text{def}}{=} \frac{\partial}{\partial \alpha} \mathbf{h}_t = \frac{\partial}{\partial \alpha} \left( \mathbf{h}_{t-1} + \alpha_{t-1}^{(h)} c_{t-1}^{-1} \left[ \delta_{t-1} \mathbf{e}_{t-1} - (\mathbf{x}_{t-1}^\top \mathbf{h}_{t-1}) \mathbf{x}_{t-1} \right] \right)$$

$$= \bar{\boldsymbol{\psi}}_{t-1} + \alpha_{t-1}^{(h)} c_{t-1}^{-1} \frac{\partial}{\partial \alpha} (\delta_{t-1} \mathbf{e}_{t-1}) - \alpha_{t-1}^{(h)} c_{t-1}^{-1} \mathbf{x}_{t-1} \mathbf{x}_{t-1}^\top \bar{\boldsymbol{\psi}}_{t-1}$$

$$= \bar{\boldsymbol{\psi}}_{t-1} - \alpha_{t-1}^{(h)} c_{t-1}^{-1} \mathbf{e}_{t-1} \mathbf{d}_{t-1}^\top \boldsymbol{\psi}_{t-1} - \alpha_{t-1}^{(h)} c_{t-1}^{-1} \mathbf{x}_{t-1} \mathbf{x}_{t-1}^\top \bar{\boldsymbol{\psi}}_{t-1}$$

where the last line follows from the fact that $\frac{\partial}{\partial \alpha} \delta_{t-1} = \mathbf{d}_{t-1}^\top \boldsymbol{\psi}_{t-1}$.

This recursive form provides a mechanism to avoid storing all previous samples and eligibility traces, and still approximate the stochastic gradient update for the stepsize. Though the above updates are exact, when implementing such a recursive form in practice, we can only obtain an estimate of $\boldsymbol{\psi}_t$, if we want to avoid storing all past data. In particular, when using $\boldsymbol{\psi}_{t-1}$ computed on the last time step $t-1$, this gradient estimate is in fact w.r.t. to the previous stepsize $\alpha_{t-2}$, rather than $\alpha_{t-1}$. Because these stepsizes are slowly changing, this gradient still provides a reasonable estimate of the actual $\boldsymbol{\psi}_{t-1}$ for the current stepsize. However, for many steps into the past, these accumulates gradients in $\boldsymbol{\psi}_t$ and $\bar{\boldsymbol{\psi}}_t$ are inaccurate. For example, even if the stepsize is nearing the optimal value, $\boldsymbol{\psi}_t$ will include larger gradients from the first step when the stepsizes where inaccurate.

To forget these outdated gradients, we maintain an exponential moving average, which focuses the accumulation of gradients in $\boldsymbol{\psi}_t$ to a more recent window. The adjusted update with forgetting parameter $0 < \beta < 1$ gives the recursive form for $\boldsymbol{\psi}_{t+1}$ and $\bar{\boldsymbol{\psi}}_{t+1}$ in (3).

## 3.2 EXTENDING TO PROXIMAL OPERATORS

A regularized GTD update for the weights can both reduce variance from noisy predictions and reduce weight on less useful features to facilitate pruning. To add regularization to GTD, for regularizer $R(\mathbf{w})$ and regularization parameter $\eta \geq 0$, we can use proximal updates (Mahadevan et al., 2014),

$$\mathbf{w}_{t+1} = \text{prox}_{\alpha \eta R}(\mathbf{w}_t + \alpha \Delta_t)$$

where $\text{prox}_{\alpha \eta R}$ is the proximal operator for function $\alpha \eta R$. The proximal operator acts like a projection, first updating the weights according to the GTD objective and then projecting the weights back to a solution that appropriately reflects the properties encoded by the regularizer. A proximal operator exists for our proposed regularizer, the clipped $\ell_2$ regularizer

$$R(\mathbf{w}) = \tfrac{1}{2} \sum_{i=1}^{d} \min(\mathbf{w}_i^2, \epsilon)$$

where $\epsilon > 0$ is the clipping threshold above which $\mathbf{w}_i$ has a fixed regularization.

Though other regularizers are possible, we select this clipped $\ell_2$ regularizer for two reasons. The clipping ensures that high magnitude weights are not prevented for being learned, and reduces bias from shrinkage. Because the predictive representation requires accurate GVF predictions, we found

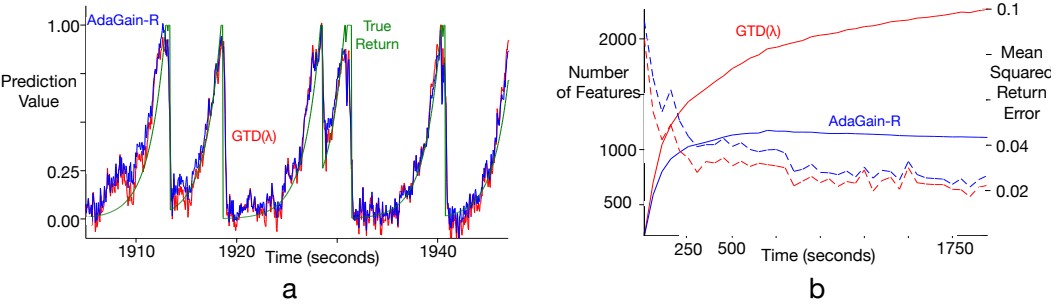

Figure 2: **(a)** An example trajectory for a single run on the robotic platform. The True Return is computed after collecting the data, and the blue and red lines corresponds to the predictions of the expected return, from that given time step. **(b)** We test how much the magnitude of the weights is spread across the features. With more spread, it becomes more difficult to identify the important features. The solid lines depict the number of features with weight magnitude above the threshold $0.001$/the number of active features. AdaGain-R spreads out values less, choosing to place higher magnitude on a subset of features. Nonetheless, AdaGain-R does not suffer in performance, as shown in the dotted lines corresponding to the y-axis on the left.

the bias without clipping prevented learning. Additionally, we chose $\ell_2$ in the clipping, rather than $\ell_1$, because the clipping already facilitates pruning, and does not introduce additional non-differentiability. The regularization below $\epsilon$ still prefers to reduce the magnitude of less useful features. For example, if two features are repeated, such a regularizer will prefer to have a higher magnitude weight on one feature, and zero weight on the other; no regularizer, or $\ell_2$ without clipping, will needlessly use both features. We provide the derivation—which closely parallels the one above—and updates in Appendix B.

## 4 USING REGULARIZATION TO PRUNE FEATURES

In this section we show that AdaGain with regularization (AdaGain-R) reduces the weights on less useful features. This investigation demonstrates the utility of the algorithm for pruning features, and so also for pruning proposed GVFs. We first test AdaGain-R on a robot platform—which is partially observable—with fixed image features. We provide additional experiments in two micro-worlds in Appendix C for pruning GVFs.

The first experiment is learning a GVF prediction on a robot, using a nonlinear expansion on input pixels. The robotic platform is a Kabuki rolling robot with an added ASUS XtionPRO RGB and Depth sensor. The agent receives a new image every 0.05 seconds and 100 random pixels are sampled to construct the state. Each pixel's RGB values are tiled with 4 tiles and 4 tilings on each colour channel, resulting in 4800 bit values. A bias bit is also included, with a value of 1 on each time step. The fixed behaviour policy is to move forward until a wall is hit and then turn for a random amount of time. The goal is to learn the value function for a policy that always goes forward; with a cumulant of 1 when the agent bumps a wall and otherwise 0; and a discount of 0.97 everywhere except when the agent bumps the wall, resulting in a discount of 0 (termination). The GVF is learned off-policy, with GTD($\lambda$) and AdaGain-R with the GTD($\lambda$) updates. Both GTD($\lambda$) and AdaGain-R receive the same experience from the behaviour policy. Results are averaged over 7 runs.

Our goal is to ascertain if AdaGain-R can learn in this environment, and if the regularization enables it to reduce magnitude on features without affecting performance. Figure 2(a) depicts a sample trajectory of the predictions made by both algorithms, after about 40k learning steps; both are able to track the return accurately. This is further emphasized in Figure 2(b), where averaged error decreases over time. Additionally, though they reach similar performance, AdaGain-R only has significant magnitude on about half of the features.

## 5 DISCOVERING GVF NETWORKS

Our approach to generating a predictive representation is simple: we generate a large collection of GVFs and iteratively refine the representation through replacing the least used GVFs with new GVFs. In the previous section, we provided evidence that our AdaGain-R algorithm effectively prunes features; we now address the larger curation framework in this section. We provide a set of GVF primitives, that enable candidates to be generated for the GVF network. We demonstrate the utility of this set, and that iteratively pruning and generating GVFs in our GVF network builds up an effective predictive representation.

### 5.1 A SIMPLE SCHEME FOR PROPOSING NEW PREDICTIONS

To enable generation of GVFs for this discovery approach, we introduce *GVF primitives*. The goal is to provide modular components that can be combined to produce different structures. For example, within neural networks, it is common to modularly swap different activation functions, such as sigmoidal or tanh activations. For networks of GVFs, we similarly need these basic units to enable definition of the structure.

We propose basic types for each component of the GVF: discount, cumulant and policy. For discounts, we consider *myopic discounts* ($\gamma = 0$), *horizon discounts* ($\gamma \in (0, 1)$) and *termination discounts* (the discount is set to 1 everywhere, except for at an event, which consists of a transition $(o, a, o')$). For cumulants, we consider *stimuli cumulants* (the cumulant is one of the observations, or inverted, where the cumulant is zero until an observation goes above a threshold) and *compositional cumulants* (the cumulant is the prediction of another GVF). We also investigate *random cumulants* (the cumulant is a random number generated from a zero-mean Gaussian with a random variance sampled from a uniform distribution); we do not expect these to be useful, but they provide a baseline. For the policies, we propose *random policies* (an action is chosen at random) and *persistent policies* (always follows one action). For example, a GVF could consist of a myopic discount, with stimuli cumulant on observation bit one and a random policy. This would correspond to predicting the first component of the observation vector on the next step, assuming a random action is taken. As another example, a GVF could consist of a termination discount, an inverted stimuli cumulant for observation one and a persistent policy with action forward. If observation can only be '0' or '1', this GVF corresponds to predicting the probability of seeing observation one changing to '0' (inactive) from '1' (active), given the agent persistently drives forward.

### 5.2 EXPERIMENTS ON DISCOVERING GVF NETWORKS IN COMPASS WORLD

We conduct experiments on our discovery approach for GVF networks in Compass World (Rafols, 2006), a partially observable grid-world where the agent can only see the colour immediately in front of it. There are four walls, with different colours; the agent observes this colour if it takes the action forward in front of the wall. Otherwise, the agent just sees white. There are five colours in total, with one wall having two colours and so more difficult to predict. The observation vector is five-dimensional, consisting of an indicator bit if the colour is observed or not. We test the performance of the learned GVF network for answering five difficult GVF predictions about the environment, that cannot be learned using only the observations. Each difficult GVF prediction corresponds to a colour, with the goal to predict the probability of seeing that colour, if the agent always goes forward. These GVFs are not used as part of the representation. A priori, it is not clear that the GVF primitives are sufficient to enable prediction in the Compass World, particularly as using just the observations in this domain enables almost no learning of these five difficult GVFs.

The GVFs for the network are generated uniformly randomly from the set of GVF primitives. Because the observations are all one bit (0 or 1), the stimuli cumulants are generated by selecting a bit index $i$ (1 to 5) and then either setting the cumulant to that observation value, $o_i$, or to the inverse of that value, $1 - o_i$. The events for termination are similarly randomly generated, with the event corresponding to a bit $o_i$ flipping. The nonlinear transformation used for this GVF network is the hyperbolic tangent. Every two million steps, the bottom 10% of the current GVFs are pruned and replaced with newly generated GVFs. Results are averaged over 10 runs.

Figure 3 demonstrates that AdaGain-R with randomly generated GVF primitives learns a GVF network—and corresponding predictive representation—that can accurately predict the five difficult

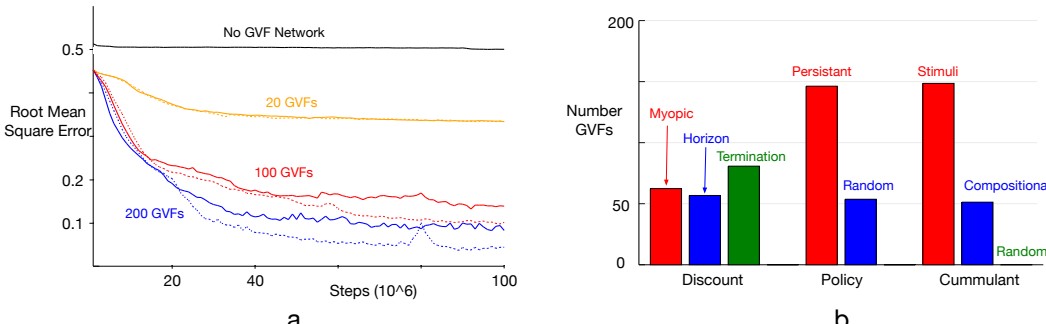

Figure 3: **(a)** Prediction error on the five difficult GVFs for variously sized networks. Solid lines correspond to networks where compositional GVFs were not generated, and dotted to networks where compositional GVFs were generated. Using only the observations (black line) results in almost no learning. **(b)** Composition of GVF network after 100 million steps of learning.

GVFs. The results show that with as few as 100 GVFs in the network, accurate predictions can be learned, though increasing to 200 is a noticeable improvement. The results also indicate that random cumulants were of no benefit, as expected, and that our system appropriately pruned those GVFs. Finally, compositional GVFs were particularly beneficial in later learning, suggesting that the system started to make better use of these compositions once the GVFs became more accurate.

## 6 DISCUSSION AND CONCLUSION

In this paper, we proposed a discovery methodology for GVF networks, to learn of predictive representations. The strategy involves iteratively generating and pruning GVF primitives for the GVF network, with a new algorithm called AdaGain to promote stability and facilitate pruning. The results demonstrate utility of this curation strategy for discovering GVF networks. There are many aspects of our system that could have been designed differently, namely in terms of the learning algorithm, generation approach and pruning approach; here, our goal was to provide a first such demonstration, with the aim to facilitate further development. We discuss the two key aspects in our system below, and potential avenues to expand along these dimensions.

In the development of our learning strategy, we underline the importance of treating the predictive representation as a dynamical system. For a standard supervised learning setting, the representation is static: the network can be queried at any time with inputs. The predictive representations considered here cannot be turned off and on, because they progressively build up accurate predictions. This dynamic nature necessitates a different view of learning. We proposed a focus on stability of the predictive system, deriving an algorithm to learn a stepsize. The stepsize can be seen as a control input, to stabilize the system, and was obtained with a relatively straightforward descent algorithm. More complex control inputs, however, could be considered. For example, a control function outputting a stepsize based on the current agent state could be much more reactive. Such an extension would the necessitate a more complex stability analysis from control theory.

Our discovery experiments reflect a life-long learning setting, where the predictive representation is slowly built up over millions of steps. This was slower than strictly necessary, because we wanted to enable convergence for each GVF network before culling. Further, the pruning strategy was simplistic, using a threshold of 10%; more compact GVF networks could likely be learned—and learned more quickly—with a more informed pruning approach. Nonetheless, even when designing learning with less conservative learning times, building such representations should be a long-term endeavour. A natural next step is to more explicitly explore scaffolding. For example, without compositional GVFs, myopic discounts were less frequently kept; this suggests initially preferring horizon and termination discounts, and increasing preference on myopic discounts once compositional GVFs are added. Further, to keep the system as simple as possible, we did not treat compositional GVFs differently when pruning. For example, there is a sudden rise in prediction error at about 80 million steps in Figure 3(b); this was likely caused by pruning a GVF whose prediction was the cumulant for a critical compositional GVF. Finally, we only considered a simple set of GVF primitives; though this simple set was quite effective, there is an opportunity to design other GVF primitives, and particularly those that might be amenable to composition.

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

# A   OTHER APPROACHES TO HANDLE PARTIAL OBSERVABILITY

There have been several approaches proposed to deal with partial observability. A common approach has been to use history: the most recent $p$ observations $\mathbf{o}_{t-1}, \ldots, \mathbf{o}_{t-p}$. For example, a blind agent in the middle of an empty room can localize itself using a history of information. Once it reaches a wall, examining its history can determine how far away it is from a wall. This could clearly fail, however, if the history is too short. Predictive approaches, like PSRs and TD networks, have been shown to overcome this issue. Further, PSRs have been shown to more compactly represent state, than POMDPs (Littman et al., 2001).

A PSR is composed of a set of action-observations sequences and the corresponding probability of these observations occurring given the sequence of actions. The goal is to find a sufficient subset of such sequences (core tests) to determine the probability of all possible observations given any action sequence. PSRs have been extended with the use of options (Wolfe and Singh, 2006), and discovery of the core tests (McCracken and Bowling, 2005). A PSR can be represented as a GVF network by using myopic $\gamma = 0$ and compositional predictions. For a test $a_1 o_1$, for example, to compute the probability of seeing $o_1$, the cumulant is 1 if $o_1$ is observed and 0 otherwise. To get a longer test, say $a_0 o_0 a_1 o_1$, a second GVF can be added which predicts the output of the first GVF (i.e., the probability of seeing $o_1$ given $a_1$ is taken), with fixed action $a_0$. This equivalence is only for computing probabilities of sequences of observations, given sequences of actions. GVF networks specify the question, not the answer, and so GVF networks do not encompass the discovery methods or other nice mathematical properties of PSRs, such as can be obtained with linear PSRs.

A TD network is similarly composed of $n$ predictions on each time step, and more heavily uses compositional questions to obtain complex predictions. Similarly to GVF networks, on each step, the predictions from the previous step and the current observations are used for this step. The targets for the nodes can be a function of the observation, and/or a function of another node (compositional). TD networks are restricted to asking questions about the outcomes from particular actions, rather than about outcomes from policies. TD networks with options (Sutton et al., 2005; Rafols, 2006) were introduced, to generalize to temporally extended actions. TD networks with options are almost equivalent to GVF networks, but have small differences due to generalizations to return specifications—in GVFs—since then. For example, options have terminating conditions, which corresponds to having a fixed discount during execution of the option and a termination discount of 0 at the end of the options. GVFs allow for more general discount functions. Additionally, TD networks, both with and without options, have a condition function. The generalization to policies, to allowing action-values to be learned rather than just value functions and the use of importance sampling corrections, encompasses these functions.

The key differences, then, between GVF networks and TD networks is in how the question networks are expressed and subsequently how they can be answered. GVF networks are less cumbersome to specify, because they use the language of GVFs. Further, once in this language, it is more straightforward to apply algorithms designed for learning GVFs. There are some algorithmic extensions to TD networks that are not encompassed by GVFs, such as TD networks with traces (Tanner and Sutton, 2005b).

# B   ADAGAIN WITH PROXIMAL OPERATORS

A proximal operator for a function $R$ with weighting $\alpha\eta$ is defined as

$$\text{prox}_{\alpha\eta R}(\mathbf{w}) \overset{\text{def}}{=} \arg\min_{\mathbf{u}} \frac{1}{2}\|\mathbf{u} - \mathbf{w}\|_2^2 + \alpha\eta R(\mathbf{u})$$

Though proximal gradient algorithsm are typically considered for convex regularizers, the proximal gradient update can be applied for our nonconvex regularizer because our proximal operator has a unique solution (Yu et al., 2015). The proximal operator for the clipped $\ell_2$ regularizer is defined element-wise, for each entry in $\mathbf{w}$:

$$\text{prox}_{\alpha\eta R}(\mathbf{w})_i = \begin{cases} \mathbf{w}_i & : \text{if } \mathbf{w}_i^2 > (1+\alpha\eta)^2 \epsilon \\ (1+\alpha\eta)^{-1}\mathbf{w}_i & : \text{otherwise} \end{cases}$$

The derivation of AdaGain with a proximal operator is similar to the derivation of AdaGain without regularization. The only difference is in the gradient of the weights, $\boldsymbol{\psi}_t = \frac{\partial}{\partial\alpha}\mathbf{w}_t$, with no change in

the gradients of $\delta_t$ and of $\mathbf{h}_t$ (i.e., $\bar{\psi}_t$). Because the proximal operator has non-differentiable points, we can only obtain a subderivative of the proximal operator w.r.t. to the stepsize. For gradient descent, we do need to more carefully use this subderivative; in practice, however, using a subderivative within the stochastic gradient descent update seems to perform well. We similarly found this to be the case, and so simply use the subderivative in our stochastic gradient descent update.

To derive the update, let $\tilde{\mathbf{w}} = \mathbf{w}_t + \alpha\Delta_t$ be the weights before applying the proximal operator. The subderivative of the proximal operator w.r.t. $\alpha$, which we call $\mathrm{dprox}_{\alpha\eta R}$, is

$$\psi_{t+1} = \mathrm{dprox}_{\alpha\eta R}(\tilde{\mathbf{w}}) \stackrel{\mathrm{def}}{=} \frac{\partial}{\partial\alpha}\mathrm{prox}_{\alpha\eta R}(\mathbf{w}_t + \alpha\Delta_t)$$

$$\mathrm{dprox}_{\alpha\eta R}(\tilde{\mathbf{w}})_i = \begin{cases} \frac{\partial}{\partial\alpha}\tilde{\mathbf{w}}_i & : \text{if } \tilde{\mathbf{w}}_i^2 > (1+\alpha\eta)^2\epsilon \\ \frac{\partial}{\partial\alpha}(1+\alpha\eta)^{-1}\tilde{\mathbf{w}}_i & : \text{otherwise} \end{cases}$$

$$= \begin{cases} \tilde{\psi}_i & : \text{if } \tilde{\mathbf{w}}_i^2 > (1+\alpha\eta)^2\epsilon \\ -(1+\alpha\eta)^{-2}\eta\tilde{\mathbf{w}}_i + (1+\alpha\eta)^{-1}\tilde{\psi}_i & : \text{otherwise} \end{cases}$$

where $\tilde{\psi}_t \stackrel{\mathrm{def}}{=} \frac{\partial}{\partial\alpha}\tilde{\mathbf{w}}_t$. The proximal operator uses $\tilde{\mathbf{w}}_{t,i}$ and $\tilde{\psi}_{t,i}$ for the element-wise update.

The resulting updates, including the exponential average with forgetting parameter $\beta$, is

$$\tilde{\mathbf{w}}_{t+1} = \mathbf{w}_t + \alpha\Delta_t$$
$$\mathbf{w}_{t+1} = \mathrm{prox}_{\alpha_t\eta R}(\tilde{\mathbf{w}}_{t+1})$$
$$\tilde{\psi}_{t+1} = ((1-\beta)\mathbf{I} - \beta\alpha_t c_t^{-1}\mathbf{e}_t\mathbf{d}_t^T)\psi_t - \beta\alpha_t c_t^{-1}\mathbf{u}_{t+1}\mathbf{e}_t^T\bar{\psi}_t + \beta\Delta_t$$
$$\bar{\psi}_{t+1} = (1-\beta)\bar{\psi}_t - \beta\alpha_t^{(h)}c_t^{-1}\left[\mathbf{e}_t\mathbf{d}_t^\top\psi_t + \mathbf{x}_t\mathbf{x}_t^\top\bar{\psi}_t\right]$$
$$\psi_{t+1} = \mathrm{dprox}_\alpha(\tilde{\mathbf{w}}_{t+1}) = \begin{cases} \tilde{\psi}_{t+1,i} & : \text{if } \tilde{\mathbf{w}}_{t+1,i}^2 > (1+\alpha\eta)^2\epsilon \\ -(1+\alpha\eta)^{-2}\eta\tilde{\mathbf{w}}_{t+1,i} + (1+\alpha\eta)^{-1}\tilde{\psi}_{t+1,i} & : \text{otherwise} \end{cases}$$

with $\psi_1 = \tilde{\psi}_1 = \bar{\psi}_1 = \mathbf{0}$.

## C  RESULTS ON PRUNING WITH ADAGAIN-R

### C.1  EXPERIMENT DETAILS FOR ROBOTIC PLATFORM

The initial step size for GTD($\lambda$) and AdaGain-R was chosen to be 0.1, with the step size of GTD($\lambda$) normalized over the number of active features (same as AdaGain-R's normalization factor). AdaGain-R has a regularization parameters $\tau = 0.001, \eta = 1$ normalized over the number of active features, and a meta-stepsize $\bar{\alpha} = 1.0$.

### C.2  PRUNING DYSFUNCTIONAL PREDICTIVE FEATURES IN CYCLE WORLD

When generating questions on the fly it is hard to know if certain questions will be learnable, or if their answers are harmful to future learning. To investigate the utility of our system for ignoring dysfunctional predictive features within the network, we conducted a small experiment in a six state cycle world domain (Tanner and Sutton, 2005a). Cycle World consists of six states where the agent progresses through the cycle deterministically. Each state has a single observation bit set to zero except for a single state with an observation of one.

We define seven GVFs for the GVF network to learn in Cycle World. Six of the GVFs correspond to the states of the cycle. The first of these GVFs has the observation bit as its cumulant, and a discount of zero. The goal for this GVF prediction is to predict the observation bit it expects to see on the next time step, which must be a '0' or a '1'. The second GVF predicts the prediction of this first GVF on the next steps: it's cumulant is the prediction from the first GVF on the next step. For example, imagine that on this step the first GVF accurately predicts a '1' will be observed on the next step. This means that on the next step, to be accurate the first GVF should predict that a '0' will be observed. Since the second GVF gets this prediction for the next time step, its target will be a '0' for this time step, and it will be attempting to predict the observation bit in two time steps. Similar, the cumulant for the third GVF is the second GVFs prediction on the next time step, and so it is aiming

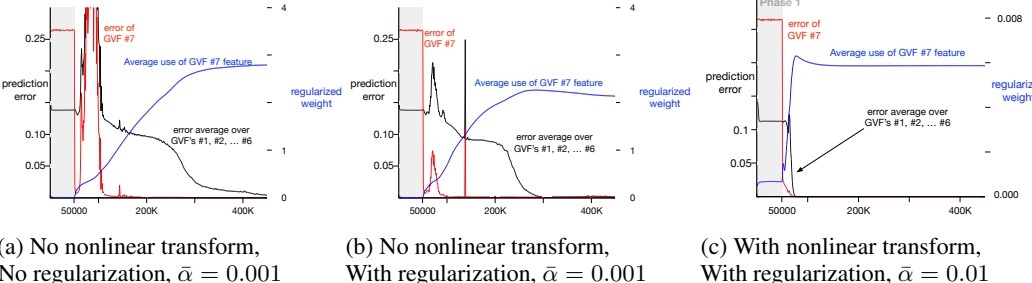

(a) No nonlinear transform, No regularization, $\bar{\alpha} = 0.001$

(b) No nonlinear transform, With regularization, $\bar{\alpha} = 0.001$

(c) With nonlinear transform, With regularization, $\bar{\alpha} = 0.01$

Figure 4: The progression of improved stability, with addition of components of our system. Even without a nonlinear transformation, AdaGain can maintain stability in the system (in **(a)**), though the meta-step size needs to be set more aggressively small. Interestingly, the addition of regularization significantly improved stability, and even convergence rate (in **(b)**). The seventh GVF is used more aggressively after Phase 1—once it becomes useful. The addition of a nonlinear transformation (in **(c)**) then finally lets the system react quickly once the seventh GVF becomes accurate. Again, without regularization, the magnitude of weights is more spread out, but otherwise the performance is almost exactly the same as (c). For both (b) and (c), the meta-parameters are $\eta = 0.1, \epsilon = 0.01$ normalized with the number of features.

to predict the observation bit in three time steps. The fourth, fifth and sixth GVFs correspondingly aim to predict the observation bit in four, five and six time steps respectively. These six GVFs all have a discount of 0, since they are predicting immediate next observations. Finally, the seventh GVF reflects the likelihood of seeing a '1' within a short horizon, where the cumulant is the observation bit on the next step and the discount is 0.9 when the observation is '0' and 0.0 when the observation is '1'. With only the first six GVFs in the GVF network, the network cannot learn to make accurate predictions (Tanner and Sutton, 2005a; Silver, 2012). By adding the seventh GVF, the whole network can reach zero error; we add this GVF, therefore, to make this domain suitable to test the algorithms. Though not the intended purpose of this experiment, it is illuminating to see the immediate benefits of the expanded flexibility of the language of GVFs beyond TD networks.

In these experiments we want to measure our system's ability to stabilize learning in a situation where a GVF in the representation is dysfunctional. To simulate the circumstance of a harmful question, we replace the seventh GVF—the critical GVF for reducing error—with random noise sampled from a Gaussian of mean and variance 0.5. When this first phase is complete, after 50k time steps, we replace the noise with the unlearned critical GVF and measure the prediction error of the system.

From Figure 4, we demonstrate that AdaGain enables stable learning under this perturbed learning setting. When the seventh GVF contained noise, AdaGain quickly dropped the step size for the others GVFs to the lower threshold $\epsilon$. This also occurred when learning without the seventh GVF, and would prevent the instability seen in Cycle World in previous work (Tanner and Sutton, 2005a; Silver, 2012). After Phase 1, once the seventh GVF begins to learn, the step-sizes are increased. In this case, the addition of a regularizer also seems to improve stability. For this experiment, there is not as clear a separation between the adapting the step-size for stability and using regularization to prune features; in fact, they both seem to play this role to some extent. The overall system, though, effectively handles this dysfunctional feature.

## C.3 PRUNING RANDOM GVFs IN COMPASS WORLD

We demonstrate the ability of our system to put higher preference on more useful GVFs in the compass world domain, and the effect of pruning less used GVFs. We construct a network with 45 expert GVFs defined in (Rafols, 2006), and 155 GVFs which produce noise sampled from a gaussian of mean 0 and variance randomly select by a uniform distribution. We prune 20 GVFs every two million steps based on the average magnitude of the feature weights, all other parameters are the same for the experiments above. Because the handcrafted expert GVFs contain useful information they should be used more by our system. Similarly to the experiments in section 5.2, we use the five evaluation GVFs to measure the representation. As we can see in figure 5, AdaGain-R does mostly remove the dysfunctional GVFs first, and when the expert GVFs are pruned the representation isn't damaged until the penultimate prune. These results also show how pruning dysfunctional or unused

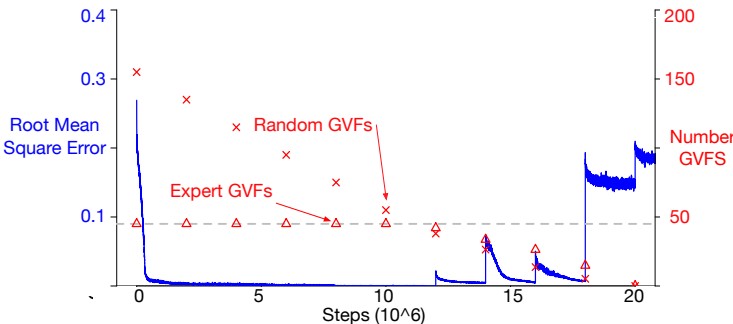

Figure 5: 45 expert GVFs and 155 random GVFs, pruning 20 GVFs every 2 million steps averaged over 10 runs.

GVFs from a representation is not harmful to the learning task. The instability seen in the ends of learning can be overcome by allowing the system to generate new GVFs to replace those that were pruned and by pruning a small amount based on the size of network used as a representation.