# OpenReview forum: "Discovery of Predictive Representations With a Network of General Value Functions"
_ICLR.cc/2018/Conference — Reject_

### Official Review · AnonReviewer1 · 2017-11-27
**not quite understand the paper**

**Rating:** 4
**Confidence:** 1

**Review:**

I have to say that I do not have all the background of this paper, and the paper is not written very clearly. I think the major contribution of the paper is represented in a very vague way.

---

> ### Author Response · Authors · 2017-12-15
> **Thank you**
>
> We would like to thank you for your time in reading our paper. While you feel you don't have much to contribute in regards to feedback, we hope you will participate in the conversation above.

---

### Official Review · AnonReviewer2 · 2017-11-27
**Thought provoking but limited**

**Rating:** 4
**Confidence:** 4

**Review:**

I really enjoyed reading this paper and stopped a few time to write down new ideas it brought up. Well written and very clear, but somewhat lacking in the experimental or theoretical results.

The formulation of AdaGain is very reminiscent of the SGA algorithm in Kushner & Yin (2003), and more generally gradient descent optimization of the learning rate is not new. The authors argue for the focus on stability over convergence, which is an interesting focus, but still I found the lack of connection with related work in this section a strange.

How would a simple RNN work for the experimental problems? The first experiment demonstrates that the regularization is using fewer features than without, which one could argue does not need to be compared with other methods to be useful. Especially when combined with Figure 5, I am convinced the regularization is doing a good job of pruning the least important GVFs. However, the results in Figure 3 have no context for us to judge the results within. Is this effective or terrible? Fast or slow? It is really hard to judge from these results. We can say that more GVFs are better, and that the compositional GVFs add to the ability to lower RMSE. But I do not think this is enough to really judge the method beyond a preliminary "looks promising".

The compositional GVFs also left me wondering: What keeps a GVF from being pruned that is depended upon by a compositional GVF? This was not obvious to me.

Also, I think comparing GVFs and AdaGain-R with an RNN approach highlights the more general question. Is it generally true that GVFs setup like this can learn to represent any value function that an RNN could have? There's an obvious benefit to this approach which is that you do not need BPTT, fantastic, but why not highlight this? The network being used is essentially a recurrent neural net, the authors restrict it and train it, not with backprop, but with TD, which is very interesting. But, I think there is not quite enough here.

Pros:
Well written, very interesting approach and ideas
Conceptually simple, should be easy to reproduce results

Cons:
AdaGain never gets analyzed or evaluated except for the evaluations of AdaGain-R.
No experimental context, we need a non-trivial baseline to compare with

---

> ### Author Response · Authors · 2017-12-15
> **Thank you**
>
> We would like to thank you for your helpful review and want to point you towards the author response thread for further discussion into the concerns you mention.

---

### Official Review · AnonReviewer3 · 2017-12-05
**Ultimately falls short of convincing**

**Rating:** 5
**Confidence:** 4

**Review:**

This paper presents a suite of algorithmic ideas to learn a network of generalized value functions.  The majority of the technical contribution is dedicated to a new algorithm, AdaGain, that adaptively selects a step size for an RL algorithm; the authors claim this algorithm has sparsifying properties.  AdaGain is used in the context of a larger algorithm designed to search for GVFs by constructing a simple grammar over GVF components.  By creating large numbers of random GVFs and then pruning away useless ones, the authors discover a state representation that is useful for prediction.

While I am deeply sympathetic to the utility and difficulty of the discovery problem in this sort of state-space modeling, this paper ultimately felt a bit weak.

On the positive side, I felt that it was well-written.  The work is well localized in the literature, and answers most questions one would naturally have.  The AdaGain algorithm is, to the best of my knowledge, novel, and the focus on "stability, not convergence" seems like an interesting idea (although ultimately, not well fleshed-out).

However, I felt that the central ideas were only thinly vetted.  For example:

* It seems that AdaGain is designed to tune a single parameter (\alpha) adaptively.  This raises several questions:
  - State-of-the-art stochastic optimizers (eg, Adam) typically introduce one step size per parameter; these are all tuned.  Why wasn't that discussed?  Would it be possible to apply something like Adam to this problem?
  - How does AdaGain compare to other adaptive gain algorithms?
  - There are ways to sparsify a representation - simple SGD + L1 regularization is a natural option.  How do we know how well AdaGain compares to this more common approach?

* The experiments seemed thin.  While I appreciated the fact that it seems that AdaGain was pruning away something, I was left wondering:
  - How generalizable are these results?  To be honest, the CompassWorld seems utterly uninteresting, and somewhat simplistic.
  - I am convinced that AdaGain is learning.  But it would be interesting to know *what* it is learning.  Do the learned GVFs capture any sort of intuitive structure in the domains?

---

> ### Author Response · Authors · 2017-12-15
> **Thank You**
>
> We would like to thank you for your helpful review and want to point you towards the author response thread for further discussion into the concerns you mention.

---

### Author Response · Authors · 2017-12-15
**Author Response and Main Thread**

We would like to thank the reviewers for their helpful comments.

The overall consensus from the reviews was that the paper was well-written, and presented interesting ideas, but that empirical results did not suggest a clear contribution, compared to existing work. We believe the confusion stems primarily because we have positioned this paper as an exploration and demonstration---not the usual case, and we will try to rectify the confusions here.

Firstly, we would like to remind the reviewers that the current state-of-the-art for predictive-question or auxiliary-task generation, is for a human designer to specify each. The goal of our paper is to provide a first investigation and successful demonstration of an autonomous discovery system---reducing the need for greatly human prior knowledge. We have provided reasonable choices for each component of our system—without suggesting that they are the best choices—to demonstrate the larger discovery framework. In this paper, we demonstrated that a discovery approach with random generation of GVFs, and a filtering strategy to prune GVFs and stabilize learning, was surprisingly effective for the discovery problem.

There are two specific concerns raised by the reviewers: the lack of justification for AdaGain and the use of micro-worlds. We admit that the relationship of AdaGain to other methods in the literature was unclear; we would like to clarify it now. Stepsize selection is an important problem in many areas, but for learned value functions in RL it is particularly problematic. Most stepsize selection strategies do not easily extend to algorithms that learn value functions, such as temporal difference (TD) algorithms. TD itself is not a gradient-based algorithm, and so it is not particularly suitable to use AdaDelta or AdaGrad. Even gradient TD (GTD) is not a standard SGD algorithm, because the gradients themselves are biased, due to the fact that the auxiliary weights only provide a (poor) estimate of a part of the gradient. We did not mean to imply that this was the first time gradient descent approaches have been used for setting stepsizes; in fact, the cited work, Benveniste (1990) provides a relatively general treatment of stochastic gradient adaptive (SGA) stepsize methods, and provides more specific algorithms for certain settings. These algorithms, and the ones cited by Sutton et al., are similar to the suggestions by Kushner and Yin; we will, however, include Kushner and Yin in the citations, for more on such approaches.

(continues in "Author Response Part 2")

---

> ### Author Response · Authors · 2017-12-15
> **Author Response Part 2**
>
> However, those approaches cannot be easily applied for the same reasons that AdaDelta and AdaGrad cannot be applied: the objectives for policy evaluation (i.e., learning value functions) make it difficult to apply standard stochastic approximation techniques. Instead, we provide a general formulation to guide the selection of the stepsize, which defines the objective to be the norm of the update. This scheme can be applied even if the update is not a gradient-descent update, and so is more suitable for the TD algorithms used in this work. Nonetheless, the suggestion to provide a baseline is a good one, and we are currently running experiments with AdaDelta, AdaGrad and Adam with GTD. Initial results with AdaDelta in CycleWorld show serious convergence issues, likely because the features change over time and the algorithm is not designed to be robust to either this change nor to application for learning value functions. We are still investigating why AdaDelta fails in this setting, but we hope to include more comprehensive results in a follow-up comment.
>
> The second concern is the simplicity of the domains used for evaluation. We chose these domains very carefully to analyze the design choices and highlight algorithm challenges. Microworlds will continue to play an important role in reinforcement learning research. Comparing algorithms on benchmark challenge problems is useful and important, but make it very difficult to understand the parts of complex agent architecture. For example, we know a set of non-trivial predictive questions which provide the agent with rough knowledge where it is in the world (approximating compass directions---hence the name). This gives a clear baseline for comparing approaches, like ours, that search the space of predictive questions---not possible in more complex domains where the set of questions is unclear. The CycleWorld and CompassWorld microworlds, were specifically designed for investigating partial observability. They may seem simplistic, but, for example, running a standard feedforward NN on these problems fails, as does any fixed history based methods. These domains have been used in several highly cited papers. Before Atari benchmarking became popular, issue-oriented research conducted in micro-worlds was one of the gold standards of scientific progress in RL. Pursuing only benchmarks has well documented limitations, and we feel that for projects like ours clear understanding is of paramount importance and best illustrated with targeted experiments.
>
> Finally, there is a comment about comparison to RNNs. We had previously avoided further expanding the scope of the paper, by motivating Predictive Representations to handle partial observability, rather than the alternative strategy of using history-based methods, such as RNNs. The Predictive Representation community is sufficiently large (c.f. PSRs, OOMs, TD-nets, TPSRs, General value functions, Auxiliary Tasks, etc.), that it warranted only investigating within that setting. Nonetheless, our next steps were to compare to RNNs, and provide more justification for Predictive Representations. We are currently running experiments with RNNs. Preliminary results indicate some issues with using RNNs, which we will report in a follow-up comment.

---

> > ### Author Response · Authors · 2018-01-03
> > **RNN Results**
> >
> > Here we provide a synopsis of the results relating to the RNN experiments as mentioned in the previous response. We will look at the cycle world domain as a first test in using RNNs to make predictions using the squared TD loss function.  The architecture we use in the experiments is an RNN using 8 GRU cells and truncated back propagation through time (BPTT) with various sequence lengths. Truncated BPTT is used for efficiency of both time and memory, which is necessary for continual learning agents. In the plots provided by an anonymous account here (https://drive.google.com/drive/folders/1n98u7_yFWgv1FL_ztCAznz8kELmQHmgD?usp=sharing) we see the state is difficult to learn when the training input doesn’t encapsulate the entire sequence of the cycle world. This is more apparent for the prediction at the current state (V(S_{t+1})), as the previous time step is more reliable. The single step look ahead prediction for the GVF network is shown as well. When running RNNs in a much larger space, a 100 state cycle world, we see similar behaviour. Learning using RNNs and truncated back propagation through time in these domains is difficult when the used input does not encapsulate the sequence in its entirety and the signal is sparse. As of now we have been unable to use RNNs to create a representation of the compass world capable of answering the evaluation questions from the paper.

---

### Decision · Program_Chairs · 2018-01-29
**ICLR 2018 Conference Acceptance Decision**

**Decision:**

Reject

**Comment:**

There was substantial disagreement between reviewers on how this paper contributes to the literature; it seems (having read the paper) that the problem tackled here is clearly quite interesting, but it is hard to tease out in the current version exactly what the contribution does to extend beyond current art.